# GazeDiff: A radiologist visual attention guided diffusion model for zero-shot disease classification

**Moinak Bhattacharya**                    MOINAK.BHATTACHARYA@STONYBROOK.EDU.EDU
**Prateek Prasanna**                        PRATEEK.PRASANNA@STONYBROOK.EDU.EDU
*Department of Biomedical Informatics, Stony Brook University, NY, US*

**Editors:** Accepted for publication at MIDL 2024

## Abstract

We present *GazeDiff*, a novel architecture that leverages radiologists' eye gaze patterns as controls to text-to-image diffusion models for zero-shot classification. Eye-gaze patterns provide important cues during the visual exploration process; existing diffusion-based models do not harness the valuable insights derived from these patterns during image interpretation. *GazeDiff* utilizes a novel expert visual attention-conditioned diffusion model to generate robust medical images. This model offers more than just image generation capabilities; the density estimates derived from the gaze-guided diffusion model can effectively improve zero-shot classification performance. We show the zero-shot classification efficacy of *GazeDiff* on four publicly available datasets for two common pulmonary disease types, namely pneumonia, and tuberculosis. Code available here.
**Keywords:** Eye-gaze, diffusion, chest x-rays, disease classification, zero-shot.

## 1. Introduction

Understanding radiologists' eye gaze patterns is crucial to deciphering the intricacies of spatial presentation of disease patterns in radiological scans. This auxiliary signal, in the form of eye gaze maps, has been recently harnessed by deep learning systems for medical image diagnosis (Bhattacharya et al., 2022b,a). Medical experts dedicate years to honing their skills in diagnosing diseases from radiology images, meticulously mastering the identification of intricate disease patterns. This experience enables their visual-cognitive working mechanism to modify/improve with time and, in turn, finesses their way of looking at scans (Bertram et al., 2016; Kelahan et al., 2019; Tourassi et al., 2013). Hence, the visual patterns of an expert can provide critical sub-visual information for a deep learning model to improve its meta-understanding of a radiology image (Stember et al., 2020).

Several works have been done in the last decade on generative modeling with a special focus on content generation. Recently, a significant improvement has been made on this front with diffusion models. Diffusion models are likelihood-based generative models that model the data distribution via an iterative noising and denoising procedure (Ho et al., 2020) and achieve state-of-the-art performance in text-based image generation. Improvements have been made in diffusion models by introducing additional controls (Zhang et al., 2023). Furthermore, conditional generative models can be converted to a classifier (Ng and Jordan, 2001) and, similarly, text-to-image diffusion models can be used as zero-shot classifiers without any additional training (Li et al., 2023a). This can be achieved by repeatedly

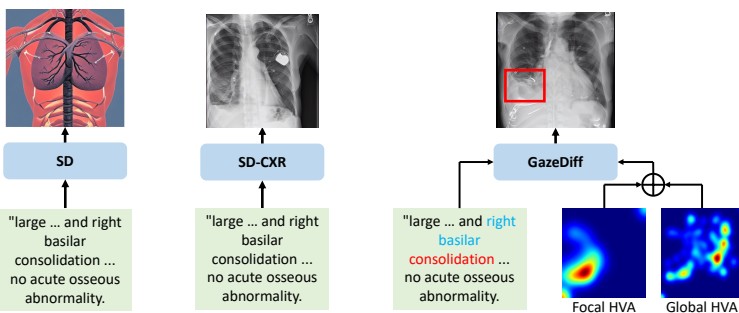

Figure 1: Three methods for generating CXRs from radiologists' transcripts. *GazeDiff* (ours) generates more clinically accurate CXRs compared to baselines.

adding noise to the input image and computing a Monte Carlo estimate of the expected noise reconstruction losses for every class in the dataset. Recent advances in controllable diffusion models enable text-to-image diffusion models with additional controls for guided image generation. Providing additional controls can be done in two ways: a) training the diffusion models from scratch (Huang et al., 2023), and b) introducing light-weight adapters into pretrained diffusion models (Zhang et al., 2023; Li et al., 2023b; Mou et al., 2023). More recently, multiple controls are also used to generate more diverse images (Zhang et al., 2023; Zhao et al., 2023a; Qin et al., 2023). Deep learning models can recognize a shape better if they can learn to generate better (Hinton, 2007). This fact goes back deep into the psychological paradigm of mechanisms to improve deep models, where generative modeling can act as a crucial player in discriminative tasks like classification. In medical image generation tasks, clinically explainable conditions and text conditions are important in generating realistic radiology images; this is still relatively unexplored.

Radiologists' eye gaze patterns are strong clinical meta-features that are highly relevant in understanding disease patterns and associated diagnoses. *Can these eye gaze patterns serve as suitable controls for diffusion models?* In this work, we propose a novel approach to integrate this expert visual attention as an additional control to the text-to-image diffusion models. Here, the text condition is the radiologist's transcript while viewing an image and contains disease-specific and context-rich information. Our proposed architecture, *GazeDiff*, utilizes these text conditions and visual attentions as additional controls for medical image generation (Figure 1).

Even though machine learning models can benefit from experts' eye gaze patterns, it is time-consuming, expensive, and often impractical to obtain eye gaze in real-time decision-making scenarios. We address this problem by adapting *GazeDiff* as a zero-shot classifier. Similar to (Li et al., 2023a), the gaze-conditioned stable diffusion model is used as a zero-shot classifier without any additional training. In our work, we show that the proposed method outperforms the baselines in classifying both known and unknown classes. In summary, we propose a novel gaze-guided zero-shot diffusion classifier, *GazeDiff*, for pulmonary disease classification.

**Motivation and Overview.** The motivation for our work stems from generating clinically

accurate medical images. We hypothesize that the context-rich visuo-cognitive information of radiologists' eye gaze patterns can be used as a clinically-relevant condition for image generation. To do this, first, we add the eye gaze patterns of experts as additional controls and radiologists' transcripts as text conditions to the text-to-image diffusion models. Then, we show that this helps in generating clinically accurate images and finally we use this finetuned diffusion model as a zero-shot classifier for downstream pulmonary diseases like pneumonia and tuberculosis classification tasks. The key contributions of this paper are as follows: a) we propose to add eye gaze patterns of experts as additional control and radiologists' text as prompt to the text-to-image diffusion models. b) we use this finetuned diffusion model as a zero-shot classifier for downstream pulmonary diseases like pneumonia and tuberculosis classification tasks.

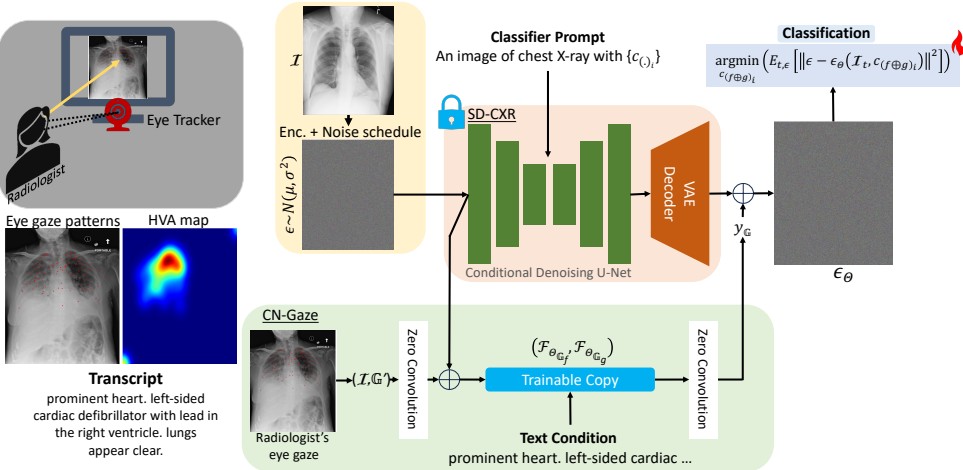

Figure 2: **Overview of *GazeDiff* architecture.** Radiologist's eye gaze patterns and the corresponding transcripts are collected. First, the Stable Diffusion block (SD-CXR) is locked and a trainable copy is created connected with zero convolution blocks to train with radiologists' eye gaze patterns as conditions. Then, a chest x-ray image $\chi$ and classifier prompt are fed to the finetuned gaze-conditioned model (CN-Gaze). This is used as a zero-shot classifier using the classifier objective.

## 2. Methodology

Figure 2 shows the pipeline on the *GazeDiff* architecture. *GazeDiff* consists of three major components: the Stable Diffusion (SD-CXR) model (Rombach et al., 2022) (shown in yellow and red), the ControlNet-Gaze (CN-Gaze model) that enhances the SD-CXR with more radiologists' eye gaze patterns as additional controls, (shown in green), and finally the class conditioned density estimates are calculated from the CN-Gaze model, (shown in blue) First, we provide a preliminary overview of diffusion models. In Section 2.1, we propose a method to use radiologists' eye gaze patterns as additional controls for text-to-image diffusion models, and in Section 2.2, we discuss a technique to use diffusion models as zero-shot classifiers.

**Preliminary.** Diffusion probabilistic models (Ho et al., 2020) or diffusion models are generative models with a parameterized Markov chain trained using variational inference. Let us consider an input image $x$, the diffusion or forward process (shown in yellow in Figure 2) is a fixed Markov process that adds Gaussian noise $\epsilon \sim \mathcal{N}(\mu, \sigma^2)$ to $x$, according to a variance schedule $\beta = \{\beta_1, ..., \beta_T\}$, shown as $q(x_{1:T}|x_0) := \prod q(x_t|x_{t-1})$, where $q(x_t|x_{t-1} := \mathcal{N}(x_t; \sqrt{1-\beta_t}x_{t-1}; \beta_t I))$. And, the reverse process(shown in red in Figure 2) is a learned Gaussian transition to denoise $x$, which can be conditioned on a variable $c$, shown as $p_\theta(x_{0:T}) := p(x_T)\prod p_\theta(x_{t-1}|x_t)$. In our case, $x$ is a CXR image, $c$ is the radiologist's findings (radiologist's transcripts and disease labels), and $T$ is the number of timesteps. So, diffusion models define $x_0$ conditioned on $c$ as $p_\theta(x_0|c) = \int_{x_{1:T}} p(x_T) \prod p_\theta(x_{t-1}|x_t, c)dx_{1:T}$ with $p_\theta(x_{t-1}|x_t) := \mathcal{N}(x_{t-1}; \mu_\theta(x_t, t), \sigma_\theta(x_t, t))$. Now, the diffusion model is trained to minimize the *variational lower bound* (ELBO) of the log-likelihood, defined as,

$$\log p_\theta(x_0|c) \geq \mathbb{E}_q \left[ \log \frac{p_\theta(x_{0:T}, c)}{q(x_{1:T}|x_0)} \right] \tag{1}$$

First, we train the SD with CXR images and we term this model SD-CXR. Then, radiologists view a CXR image $\mathcal{I}$ and generate eye gaze patterns $\mathbb{G}$, discussed in detail in 2.1. Similar to (Bhattacharya et al., 2022b,a), Human Visual Attention(HVA) maps are computed from $\mathbb{G}$. In this work, we compute separate Focal HVA and Global HVA maps. The focal HVA captures fine-grained disease-relevant features while the global HVA captures coarse disease-relevant features; a weighted combination of these maps captures the entire feature space of disease-relevant regions, discussed in detail in Appendix B.

### 2.1. Gaze as an additional control for Text-to-Image diffusion

Here, we discuss CN-Gaze, in which the radiologists' eye gaze patterns are injected as additional conditions into the SD-CXR model. Let us assume that the SD-CXR model is $\mathcal{F}(.)$. We represent radiologists' eye gaze patterns as $\mathbb{G}_r \in R^{(\mathbb{F}, \mathcal{T})}$, where $r \in \{1, 2, ..., \mathcal{R}\}$. Here, $\mathbb{F}$ are the eye gaze fixations over time $\mathcal{T}$ and $\mathcal{R}$ is the number of radiologists whose eye gaze are collected for CXR image $\mathcal{I} \in \mathbb{R}^{H \times W \times C}$ with $\{H, W, C\}$ are height, width, and number of channels of the image. Now, HVA edge maps are computed for Global and Focal HVA from $\mathbb{G}$, represented as $\mathbb{G}'$ (discussed in detail in the Appendix B). Similar to (Zhang et al., 2023), for training with additional control $\mathbb{G}'$, we freeze $\mathcal{F}(.)$ with initial parameters represented as $\Theta$ and clone the frozen model parameters into a *trainable* model to train with the gaze condition, shown as $\Theta'_\mathbb{G}$. The input $\mathcal{I}$ is fed to both $\mathcal{F}_\Theta(.)$ and $\mathcal{F}_{\Theta'_\mathbb{G}}(.)$ in a manner where $\mathcal{F}_{\Theta'_\mathbb{G}}$ blocks are connected to $\mathcal{F}_\Theta$ blocks through *zero convolution*(represented as $\mathbb{Z}(.)$, which is a $Conv_{1 \times 1}$ layer with $W = 0$ and $b = 0$) layers, as shown in Figure 2. The output $y'_\mathbb{G}$ is shown as $y_\mathbb{G} = \mathcal{F}_\Theta(\mathcal{I}) + \mathbb{Z}_2(\mathcal{F}_{\Theta_\mathbb{G}}(\mathcal{I} + \mathbb{Z}_1(\mathbb{G})))$. In our case, the radiologist's eye gaze patterns are represented as two separate entities, namely, Focal HVA and Global HVA. Hence, two separate ControlNets are trained, $\mathcal{F}_{\Theta_f}$, and $\mathcal{F}_{\Theta_g}$. The resulting outputs from these ControlNets are added with no extra weighting or linear interpolation to make it a Multi-ControlNet $\mathcal{F}_{\Theta_f}$. Hence, $y'_\mathbb{G}$ can be represented as a weighted combination of focal-conditioned, $y_f$ and global-conditioned, $y_g$, shown as,

$$y_\mathbb{G} = \lambda_1 y_f + \lambda_2 y_g, \lambda_1, \lambda_2 \in \mathbb{R}^+, \left.\begin{array}{l} y_f = \mathcal{F}_\Theta^1(\mathcal{I}) + \mathbb{Z}_2^1(\mathcal{F}_{\Theta_{\mathbb{G}_f}}^1(\mathcal{I} + \mathbb{Z}_1^1(\mathbb{G}_f))) \\ y_g = \mathcal{F}_\Theta^2(\mathcal{I}) + \mathbb{Z}_2^2(\mathcal{F}_{\Theta_{\mathbb{G}_g}}^2(\mathcal{I} + \mathbb{Z}_1^2(\mathbb{G}_g))) \end{array}\right\} \text{HVA} \tag{2}$$

## 2.2. Zero-Shot classification

In common medical scenarios, during real-time inference, radiologists' eye gaze patterns and transcripts are not available. Here we discuss how CN-Gaze is used for for zero-shot classification. Given each noised sample $x_t = \sqrt{\alpha_t}x + \sqrt{1-\alpha_t}\epsilon$, diffusion model learns $\epsilon_\theta(x_t, c)$. Using this parameterization, Equation (1) can be rewritten as, $-\mathbb{E}[\sum_{t=2}^{T} w_t\|\epsilon - \epsilon_\theta(x_t, c)\|^2 - \log p_\theta(x_0|x_1, c)] + C$. From (Li et al., 2023a), assuming $w_t = 1$ and $\log p_\theta(x_0|x_1, c) \approx 0$ as $T = 1000$ is large, the simplified ELBO term is represented as $-\mathbb{E}_{t,\epsilon}\left[\|\epsilon - \epsilon_\theta(\mathcal{I}_t, c)\|^2\right] + C$. Now, classification tasks using generative models can be defined using Bayes Theorem as $p_\theta(c_i|\mathcal{I}) = \frac{p(c_i)p_\theta(\mathcal{I}|c_i)}{\sum_j p(c_j)p_\theta(\mathcal{I}|c_j)}$, where $c_i$ is the label, and $\mathcal{I}$ is the input image. Using the simplified ELBO term, it can be re-written as $p_\theta(c_i|\mathcal{I}) = \frac{exp\left(-\mathbb{E}_{t,\epsilon}\left[\|\epsilon-\epsilon_\theta(\mathcal{I}_t, c_i)\|^2\right]\right)}{\sum_j exp\left(-\mathbb{E}_{t,\epsilon}\left[\|\epsilon-\epsilon_\theta(\mathcal{I}_t, c_j)\|^2\right]\right)}$. In our case, from Equation (2), $p_\theta(c_i|\mathcal{I})$ can be rewritten as $p_\Theta(c_{(f\oplus g)_i}|\mathcal{I})$, shown as

$$p_\Theta(c_{(f\oplus g)_i}|\mathcal{I}) = \frac{exp\left(-\mathbb{E}_{t,\epsilon}\left[\left\|\epsilon - \epsilon_{\Theta_{(f\oplus g)}}(\mathcal{I}_t, c_{(f\oplus g)_i})\right\|^2\right]\right)}{\sum_j exp\left(-\mathbb{E}_{t,\epsilon}\left[\left\|\epsilon - \epsilon_{\Theta_{(f\oplus g)}}(\mathcal{I}_t, c_{(f\oplus g)_j})\right\|^2\right]\right)} \tag{3}$$

Then, an unbiased Monte Carlo estimate is calculated for each expectation by sampling $N(t_i, \epsilon_i)$, shown as $\frac{1}{N}\sum_{i=1}^{N}\left\|\epsilon_i - \epsilon_\Theta(\sqrt{\alpha_t}\mathcal{I} + \sqrt{1-\alpha_t}\epsilon_i, c_j)\right\|^2$. Now, plugging this formulation into Equation (3) makes the zero-shot *GazeDiff* classifier.

## 3. Experiments and Results

### 3.1. Datasets and Experiments

We use the radiologist's eye gaze and corresponding transcriptions from publicly available Eye Gaze Data for Chest X-rays (Karargyris et al., 2021, 2020) (n=1083). For zero-shot classification, we show results on two pneumonia classification and two tuberculosis classification datasets. For pneumonia classification, we use the test set of publicly available Cell Pneumonia dataset (Kermany et al., 2018) and RSNA Pneumonia Detection challenge dataset (Shih et al., 2019). For tuberculosis classification, we use the NLM MCU (Candemir et al., 2013; Jaeger et al., 2013, 2014) and CHN (Jaeger et al., 2014) dataset which are obtained from Montgomery County, Maryland, USA and Shenzhen No. 3 People's Hospital in China, respectively. The REFLACX dataset (Bigolin Lanfredi et al., 2022; Lanfredi et al., 2021) is used for evaluating the quality of the generated images.

For finetuning, we train the SD v1.5 for 15000 steps with image size of $512 \times 512$ on 1 Quadro RTX 8000 (48 GB) with a batch size of 4 and a learning rate of $1e-5$. With this finetuned SD as the base, we train a ControlNet model with HVA edge maps. Here we compute Canny edges (Canny, 1986) of the global and focal HVA maps separately, as shown in Figure 3. This training was performed for 10 epochs with a batch size of 2 and a learning rate of $1e-5$. During inference, we merge the global and focal ControlNets with UniPCMultistepScheduler (Zhao et al., 2023b) for sampling. The number of time steps is set to 50 and the condition scale is set to 2.5. Also, we use $\ell_2$ norm for $\epsilon$-prediction error.

### 3.2. Quantitative Results

We show results on the known class (samples from this class present in training SD-CXR and CN-Gaze), i.e. pneumonia classification, and the unknown class (samples from this class not present in training SD-CXR and CN-Gaze), i.e. tuberculosis classification.

Table 1: **Zero-Shot Classification performance.** We experiment with 4 publicly available datasets for the known class (shown as ✓) and unknown class (shown as ✗).

| Pathology | CXR | Pneumonia | | | | Tuberculosis | | | |
|---|---|---|---|---|---|---|---|---|---|
| Dataset | - | Cell✓ | | RSNA✓ | | CHN✗ | | MCU✗ | |
| Metrics | - | Acc.(↑) | F1(↑) | Acc.(↑) | F1(↑) | Acc.(↑) | F1(↑) | Acc.(↑) | F1(↑) |
| SD | ✗ | 52.24 | 49.49 | 49.23 | 32.67 | 43.96 | 10.60 | 52.90 | 42.20 |
| CN | ✗ | 50.32 | 47.81 | 49.30 | 32.70 | 46.88 | 24.14 | 54.35 | 43.98 |
| CLIP | ✗ | 50.16 | 60.38 | - | - | 50.45 | 46.92 | 52.90 | **62.43** |
| SD-CXR | ✓ | 58.49 | 71.93 | 49.30 | 38.89 | 58.61 | 70.34 | **57.25** | 56.22 |
| CN-CXR | ✓ | 58.01 | 71.21 | 49.27 | 39.07 | 58.61 | 70.41 | 56.52 | 55.12 |
| PubMedCLIP | ✓ | 23.56 | 21.93 | 19.88 | 28.57 | 56.04 | 63.21 | **57.25** | 47.79 |
| RoentGen | ✓ | 54.01 | 62.29 | 41.10 | 34.07 | 50.15 | 58.12 | 44.93 | 44.78 |
| GazeDiff(Ours) | ✓ | **59.29** | **72.08** | **49.60** | **39.31** | **58.91** | **70.69** | **57.25** | 56.00 |

**Evaluation of Zero-shot classification Performance.** *GazeDiff* is compared against standard diffusion models like SD(Ho et al., 2020), ControlNet(Zhang et al., 2023) trained on natural images, and the same models finetuned on CXRs (i.e. SD-CXR and CN-CXR). We also evaluate the performance of *GazeDiff* against RoentGen(Chambon et al., 2022), a baseline SD model finetuned with CXR images. To measure the classification performance, we report Accuracy(↑) and F1-score(↑) in Table 1, and we report additional metrics in Appendix Table 4. *GazeDiff* outperforms the baselines on all 4 benchmark datasets for Pneumonia and Tuberculosis classification. We observe that *GazeDiff* outperforms the finetuned SD model by **0.62±0.48%** [Cell: 1.35%, RSNA: 0.60%, CHN: 0.51%, MCU: *no improvement*], the finetuned ControlNet model by **1.15±0.65%** [Cell: 2.16%, RSNA: 0.66%, CHN: 0.51%, MCU: 1.27%] and RoentGen by **15.60±4.55%** [Cell: 8.90%, RSNA: 17.13%, CHN: 14.87%, MCU: 21.51%]. We also show comparisons with CLIP(Radford et al., 2021) and PubMedCLIP(Eslami et al., 2021).

Table 2: Quantitative assessment of image fidelity and diversity on REFLACX dataset. FID, and CLIP-score for different models are reported.

| REFLACX | CN-CXR | | GazeDiff | |
|---|---|---|---|---|
| - | FID(↓) | CLIP(↑) | FID(↓) | CLIP(↑) |
| Atelectasis | 462.75 | 24.65 | 420.05 | 24.73 |
| Consolidation | 449.38 | 23.69 | 409.13 | 23.87 |
| Edema | 442.91 | 25.75 | 418.34 | 25.96 |
| Emphysema | 413.33 | 27.32 | 363.04 | 27.26 |

**Evaluation of image quality**. In Table 2, we report FID(↓), and CLIP-score(↑) to evaluate the performance of *GazeDiff* for generated images quality and compare it with ControlNet. We show that *GazeDiff* outperforms ControlNet on 4 pulmonary disease types. Additional results are reported in Appendix Table 5 and Table 6.

### 3.3. Ablation Analysis

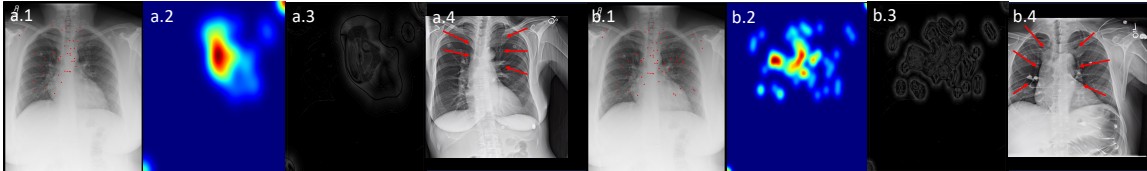

Figure 3: **Ablation Analysis.** (a.*) images for focal HVA computations. (b.*) images for global HVA computations. (*.1) raw fixations overlayed on the CXR. (*.2) the HVA map. (*.3) the canny edge map. (*.4) the *GazeDiff* generated CXR. The red arrows show disease patterns generated in the HVA regions.

In Table 3, we show the performance of *GazeDiff*, when trained with different human visual attentions. From a radiologist's eye gaze patterns, we calculated two different visual attention patterns, namely global attention and focal attention, described in detail in Appendix B. Here, we observe that the ControlNet finetuned with the combined global and focal attention mechanisms generate better noise representation for zero-shot classifications and outperform the Global model by **1.35%** and the Focal model by **1.62%**. In

Table 3: **Ablation Results.** We compare *GazeDiff*, which has global and focal human visual attention maps as muli-controls, with focal-only and global-only control models. Results reported on the Cell Pneumonia dataset.

| Cell | Accuracy(↑) | F1(↑) | Precision(↑) | Recall(↑) |
|------|-------------|-------|--------------|-----------|
| Focal | 58.33 | 71.49 | 62.45 | 83.59 |
| Global | 58.49 | 71.57 | 62.57 | 83.59 |
| Global+Focal | **59.29** | **72.09** | **63.08** | **84.10** |

Figure 3, we show the generated CXRs for the different human visual attention canny edge maps. Here, we observe that the generated images show distinct irregularities in locations where there are canny edges of the human visual attentions. This demonstrates the robust interpretation of the experts' eye gaze content semantics for medical image generation.

### 3.4. Qualitative Results

In Figure 5, we compare the generated CXRs from *GazeDiff* with different baselines like Stable Diffusion, ControlNet, and RoentGen for pneumonia and tuberculosis disease names as text conditions. Here, we observe that the *GazeDiff* generates more realistic disease patterns when compared to the baselines. We also show the location of the generated

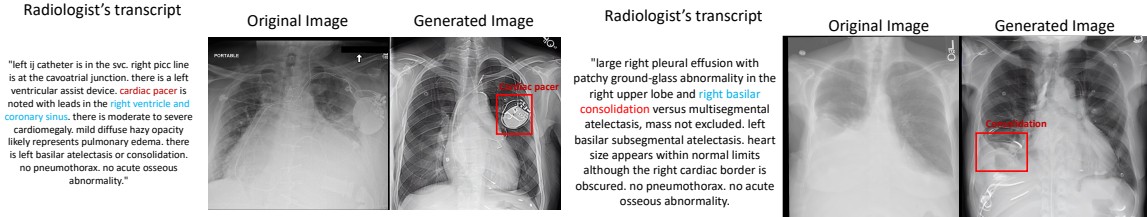

Figure 4: **Qualitative Results.** We show the CXRs generated by *GazeDiff* based on the radiologists' transcript as text conditions. We show the generated pathology/object mentioned in the radiologists' transcript in red box.

disease patterns annotated by a radiologist (7 years experience). In Figure 4, we show the generated CXRs of the proposed method for different transcriptions as text conditions. We show the location of the generated disease patterns annotated in red. *GazeDiff* not only generates disease patterns/irregularities and devices (text highlighted in red) as mentioned in the transcript but also generates them in the mentioned location (text highlighted in blue).

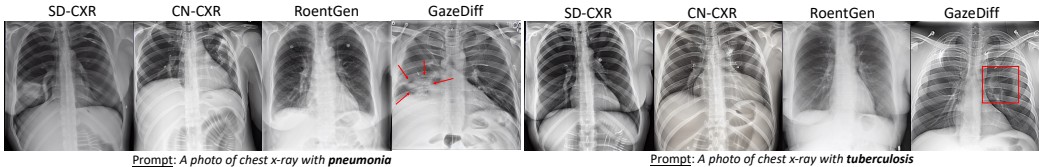

Figure 5: **Qualitative Comparisons.** We compare the CXRs generated by *GazeDiff* with baselines based on a class-conditioned prompt. red arrows/bounding box show the generated pathology.

## 4. Conclusion

In this work, we show that eye gaze patterns are indeed critical in generating high-fidelity and robust medical images. With eye gaze patterns as conditional inputs to diffusion models, the generated medical images contain disease-specific features or irregularities in the exact location or periphery of radiologists' eye gaze fixations. This makes the medical image generation more robust and disease-specific, possibly minimizing hallucinations. In another important contribution of our work, we show that the diffusion model trained with eye gaze patterns can also be used as a zero-shot classifier for two common disease types without any additional training. This work is a significant step forward in gaze-conditioned medical discriminative tasks using a generative modeling approach. However, there is a scope for improvement in zero-shot classification performance by selecting prompts that are more gaze-relevant and hence induce more diversity in the image generation step.

## Acknowledgments

The reported research was partly supported by NIH 1R21CA258493-01A1, NIH 75N92020D00021 (subcontract), and the OVPR and IEDM seed grants at Stony Brook University. The content is solely the respon- sibility of the authors and does not necessarily represent the official views of the National Institutes of Health.

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

# Appendix A. Datasets

In this work, we use 4 publicly available datasets. Here, we provide detailed description of the datasets. We use Eye Gaze Data for Chest X-rays (Karargyris et al., 2021) (n=1083) and for evaluation use REFLACX dataset(Bigolin Lanfredi et al., 2022; Lanfredi et al., 2021). For zero-shot classification, we use **Cell Pneumonia Classification**: In this dataset, the CXRs were selected from pediatric patients of 1-5 years old from Guangzhou Women and Children's Medical Center, Guangzhou. We use the test set that consists of 234 normal CXR images and 390 pneumonia (viral and bacterial) CXR images. **RSNA Pneumonia Detection Challenge Dataset** This dataset was made public by the Radiological Society of North America (RSNA). We have used the test set that consists of 4527 CXR images. **China** This dataset was obtained from Shenzhen No. 3 People's Hospital in China. We use the test set that has 284 normal CXRs and 378 CXRs with tuberculosis. **Montgomery County dataset** This dataset was acquired from the Department of Health and Human Services, Montgomery County, Maryland, USA. We use the test set that has 80 normal CXRs and 58 CXRs with tuberculosis. In Figure 6, we show images containing different pathologies from these mentioned datasets.

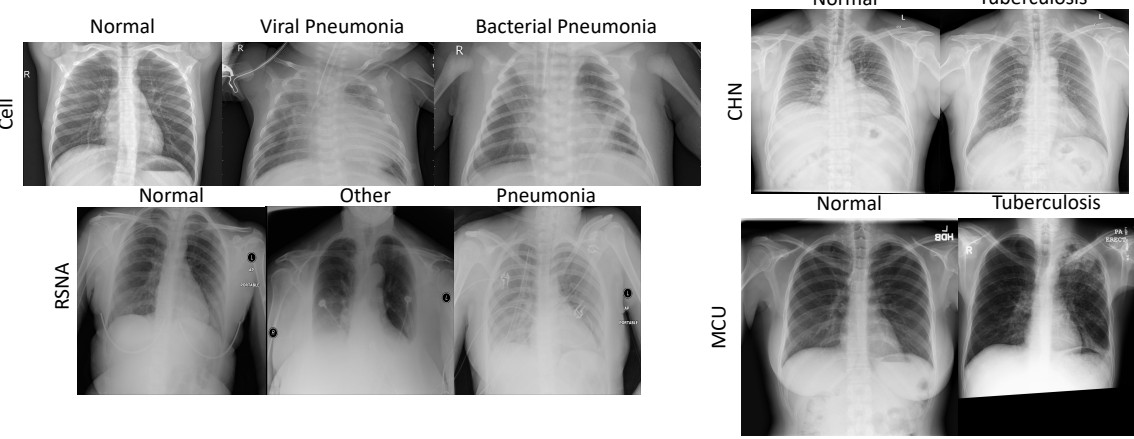

Figure 6: **Dataset description.** The CXR images from different datasets with different disease are shown.

# Appendix B. Human Visual Attention (HVA) computation

**Focal HVA.** Given radiologists' eye gaze patterns $\mathbb{G}_r$, for a radiologist $i$, the fixation points set is represented as $g_i \in \mathbb{R}^{\mathbb{G}_i}$. For focal HVA calculation, we select the cluster of points $\mathcal{C}_k^I \subset g_i$, where $k \in \{1, 2, ..., K\}$ is the total number of random clusters, such that $\forall (x_k, y_k) \in \mathcal{C}_k^I : \|x_k - y_k\|_{\mathcal{D}} \leq \|x_j - y_j\|_{\mathcal{D}}$, here $k \cap j = \emptyset$, shown in Figure 7.

**Global HVA.** Similar to Focal HVA, in Global HVA calculation, we select the cluster of points $\mathcal{C}_k^D \subset g_i$, such that $\forall (x_k, y_k) \in \mathcal{C}_k^D : \|x_k - y_k\|_{\mathcal{D}} \geq \mathbf{c}$, where $\mathbf{c} \in \mathbb{R}$. Then, a multi-

dimensional Gaussian filter with standard deviation, $\sigma = 64$, is used to generate these attention heatmaps, shown in Figure 7.

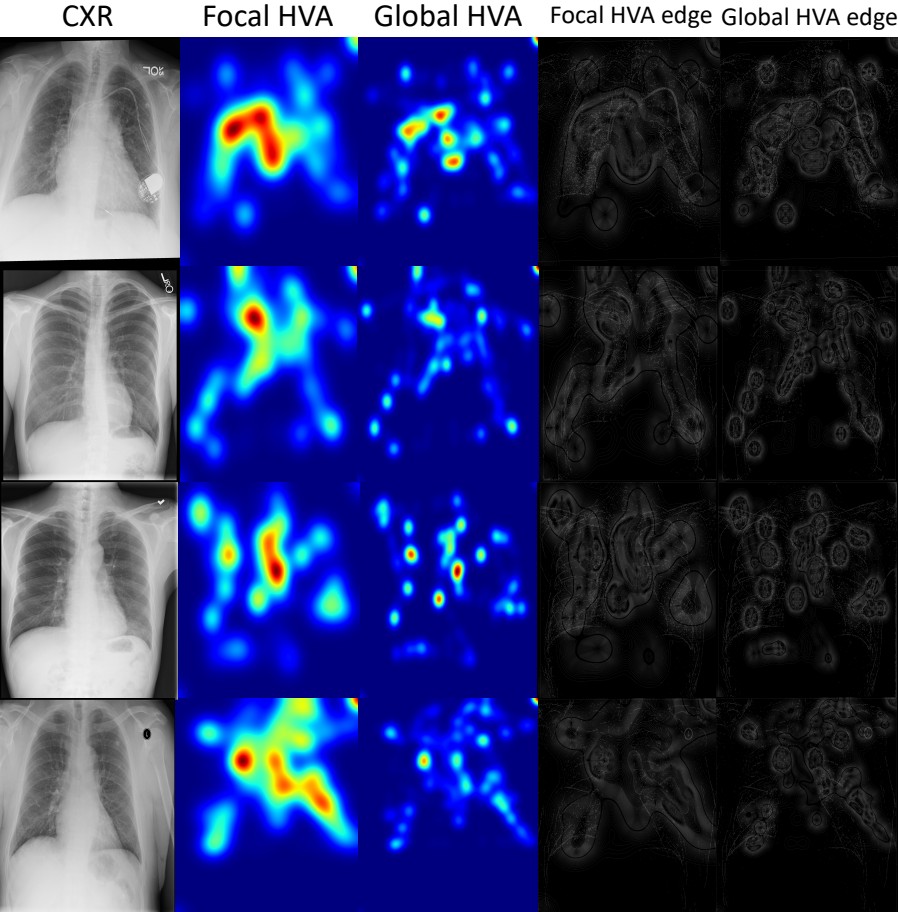

Figure 7: **HVA computation.** The HVA maps along with the HVA edge maps are shown.

## Appendix C. Additional quantitative results

Table 4: **Additional performance metrics for Zero-Shot Classification performance.** We report Precision and Recall on 4 publicly available datasets for the Pneumonia and Tuberculosis disease classification.

| Pathology | Pneumonia | | | | Tuberculosis | | | |
|---|---|---|---|---|---|---|---|---|
| Dataset(Known) | Cell✓ | | RSNA✓ | | CHN✗ | | MCU✗ | |
| Metrics | Prec. | Recall | Prec. | Recall | Prec. | Recall | Prec. | Recall |
| SD | 73.00 | 37.44 | 34.62 | 49.23 | 59.46 | 5.82 | 40.05 | 52.90 |
| SD+ft. | 62.29 | **85.13** | 39.88 | 49.30 | 59.52 | 85.98 | 63.60 | **57.25** |
| CN | **69.60** | 36.41 | 34.64 | 49.30 | **65.12** | 14.81 | 44.33 | 54.35 |
| CN+ft. | 62.31 | 83.08 | **42.08** | 49.27 | 59.49 | 86.24 | 63.36 | 56.52 |
| RoentGen | 63.88 | 60.77 | 33.35 | 41.10 | 55.85 | 60.58 | 44.66 | 44.93 |
| GazeDiff(Ours) | 63.08 | 84.10 | 40.28 | **49.60** | 59.64 | **86.77** | **63.93** | **57.25** |

Table 5: Additional Quantitative assessment of image fidelity and diversity on REFLACX dataset. FID, and CLIP-score for different models are reported.

| REFLACX | CN-CXR | | GazeDiff | |
|---|---|---|---|---|
| - | FID($\downarrow$) | CLIP($\uparrow$) | FID($\downarrow$) | CLIP($\uparrow$) |
| Fracture | 426.51 | 16.90 | 395.18 | 17.65 |
| Mass | 455.46 | 25.38 | 406.63 | 25.29 |
| Opacity | 411.44 | 24.66 | 388.91 | 24.83 |
| Pleural Abnormality | 471.71 | 23.19 | 403.63 | 23.28 |
| Pneumothorax | 433.44 | 23.91 | 385.39 | 24.03 |
| Support Devices | 442.24 | 23.84 | 392.71 | 23.80 |

Table 6: Quantitative assessment on REFLACX dataset for SD-CXR and RoentGen. FID, and CLIP-score for different models are reported.

| REFLACX | SD-CXR | | RoentGen | |
|---|---|---|---|---|
| - | **FID($\downarrow$)** | **CLIP($\uparrow$)** | **FID($\downarrow$)** | **CLIP($\uparrow$)** |
| Atelectasis | 439.20 | 24.96 | 440.34 | 25.20 |
| Consolidation | 412.60 | 24.20 | 439.53 | 24.32 |
| Edema | 417.12 | 25.54 | 436.25 | 25.90 |
| Emphysema | 384.11 | 27.26 | 370.93 | 27.43 |
| Fracture | 391.15 | 17.46 | 388.72 | 17.69 |
| Mass | 402.47 | 25.49 | 426.91 | 26.43 |
| Opacity | 377.59 | 25.58 | 379.95 | 25.02 |
| Pleural Abnormality | 414.51 | 23.59 | 432.01 | 23.69 |
| Pneumothorax | 402.71 | 23.79 | 397.07 | 23.84 |
| Support Devices | 396.25 | 24.01 | 405.79 | 24.18 |

Table 7: Ablation on additional conditions on Cell and CHN datasets. Accuracy, and F1-score are reported for different conditions.

| REFLACX | Cell | | CHN | |
|---|---|---|---|---|
| - | **Acc.($\uparrow$)** | **F1($\uparrow$)** | **Acc.($\uparrow$)** | **F1($\uparrow$)** |
| Canny | 58.01 | 71.21 | 58.61 | 70.41 |
| Sobel | 58.65 | 71.71 | 58.31 | 70.06 |
| GL | 58.49 | 71.57 | 58.76 | 70.49 |
| Segmentation | 58.65 | 71.77 | 58.91 | 70.63 |
| GazeDiff (Ours) | **59.29** | **72.08** | **58.91** | **70.69** |

## Appendix D. Additional figures

Stable Diffusion    ControlNet         SD-CXR         CN-CXR

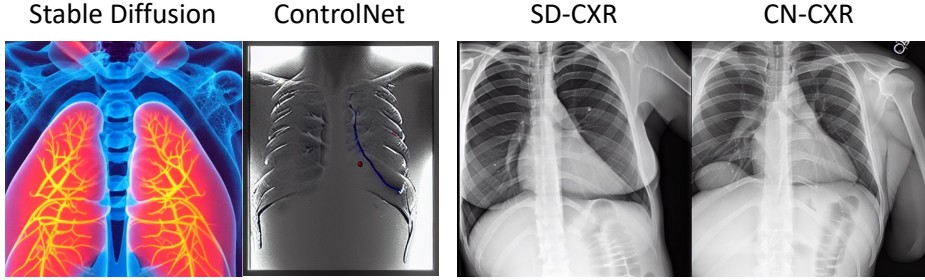

Prompt: *A chest xray with pneumonia*

Figure 8: **Finetuning diffusion models.** We show that the generated CXRs for Stable Diffusion and ControlNet models without finetuning look unrealistic, whereas after finetuning the x-rays look realistic for the mentioned prompt.

Original Image          Generated Image

Radiologist's transcript

"no pneumothorax. cardiac silhouette normal. clear lungs."

Figure 9: **Qualitative Result-Normal** We show a normal CXR generated by *GazeDiff* based on the radiologist's transcript as text condition.

