# OpenReview forum: "GazeDiff: A radiologist visual attention guided diffusion model for zero-shot disease classification"
_MIDL.io/2024/Conference — MIDL 2024 Poster_

### Official Review · Reviewer_hvVX · 2024-02-25

**Confidence:** 3
**Preliminary Rating:** 2
**Final Rating:** 3.5

**Summary:**

In this work, the authors proposed a method named GazeDiff which aims to utilize radiologists' eye gaze patterns to improve the classification results on chest X-ray diseases. This work uses the framework of stable diffusion as the backbone of the diffusion model to perform disease classification in a zero-shot manner. In details, the authors finetuned the stable diffusion network based on chest X-ray from a public dataset with available eye gaze data and conditioned the diffusion model with processed eye gaze data, actually HVA images, to improve the performance. This conditioned diffusion model follows previously proposed method where the diffusion model is used as a classifier given different class labels. The authors chose to use ControlNet as the condition method where the processed eye gaze patterns are used as input with zero-init conv. The work achieves relatively good performance compared with the baselines (stable diffusion and stable diffusion + controlNet).

**Strengths:**

1. This work uses additional supporting information from eye gaze data which is interesting.

2. The paper is pretty well organized and written.

3. Choosing ControlNet as the condition method is appropriate.

**Weaknesses:**

1. The novelty is limited. This work basically uses diffusion model as a classifier and chooses ControlNet as the condition method which is quite common.

2. Lack of comparisons with other X-ray disease classification model. There are many works related to this domain. It would be better if the authors could add more baselines from other X-ray disease classification model to demonstrate the superiority.

3. The performance doesn't clearly outperform other baselines such as CN-CXR. The metrics are very close.

4. The overall writing is good but the details or descriptions for baselines are not clearly defined. I understand it might be due to the 8 page limits.

**Detailed Comments:**

1. The abbreviation of the name for ControlNet (CN) is not clearly stated. It would be better to clearly define somewhere in your paper. Same for RoentGen. A reference in the paper or appendix is sufficient.

2. What is the input of ControlNet for baseline? What's the condition image is it?

3. SD and CN is not really necessary for the results since they are all trained with large dataset including natural imaging not specifically with X-ray image.

**Justification Of Final Rating:**

Even though I think the novelty of the paper is still limited. But I'm satisfied with the rebuttals of the other concerns. After the rebuttal, the paper now is rich and well-written. I think it's improved in other aspects. I'm changing my rating from weak reject to borderline accept.

**Justification Of The Preliminary Rating:**

The main part is the novelty as stated in the weakness section. The idea is pretty cool that utilizing eye gaze data as supporting information but I think the experiments are not sufficient. At least other approaches for X-ray disease classification should be compared.

**Questions To Address In The Rebuttal:**

1. Finetuning of Stable Diffusion: How is stable diffusion fine-tuned, e.g., how do the authors handle one-channel X-ray input; Is the text module also finetuned?

2. What's the input of the copied and trained ControlNet compared with your proposed GazeDiff since ControlNet also requires an image or 2D input.

3. Experiments using only canny edge of the X-ray images instead of HVA. How would the performance of this be?

4. Experiments of other condition methods if time allows.

---

> ### Author Response · Authors · 2024-03-17
>
> We thank the reviewer for the excellent comments on our work. We answer the questions/comments below:
>
> > The novelty is limited. This work basically uses diffusion model as a classifier and chooses ControlNet as the condition method which is quite common.
>
> The reviewer expresses concern that using ControlNet is common and hence the novelty is limited. We would like to point out that we are not just using ControlNet as the conditioning method for downstream zero-shot classification. We train the ControlNet using HVA maps that contain the radiologist’s eye gaze attention patterns for downstream zero-shot classification. This approach is still unexplored and opens up avenues for future method developments.
>
> > Lack of comparisons with other X-ray disease classification model. There are many works related to this domain. It would be better if the authors could add more baselines from other X-ray disease classification model to demonstrate the superiority.
>
> There are very few existing methods that demonstrate zero-shot classification with CXRs. In Table 1 of [2], SD and CLIP are used for comparison. Hence, we also add CLIP[3] and PubMedCLIP[4] as a baseline method in this rebuttal (see Table 1).
>
> > The performance doesn't clearly outperform other baselines such as CN-CXR. The metrics are very close.
>
> Although the metrics are close, we show statistical significance of the method by calculating the paired t-test between the predictor errors ($\epsilon_\theta$ in Figure 1) for 2 datasets. For the Cell Pneumonia dataset, the p value is 0.0067 and for the RSNA Pneumonia dataset, the p value is 0.0317.
>
> > The abbreviation of the name for ControlNet (CN) is not clearly stated. It would be better to clearly define somewhere in your paper. Same for RoentGen. A reference in the paper or appendix is sufficient.
>
> Thank you very much for pointing out this inconsistency. We modify the first line of Section 2 where we define that SD is Stable Diffusion and CN is ControlNet. The name of the RoentGen[1] paper is RoentGen and we are using the same name everywhere in the paper. For better understanding, we add the references for the baselines in Subsection 3.2.
>
> > SD and CN is not really necessary for the results since they are all trained with large dataset including natural imaging not specifically with X-ray image.
>
> The author rightly points out that SD and CN are trained with large natural imaging datasets. We wanted to compare that the performance of these standard SD and CN features are not better than SD and CN finetuned on CXRs. We modify Table 1 and add another column ‘CXR’ that shows models trained with CXRs or natural images.
>
> > Finetuning of Stable Diffusion: How is stable diffusion fine-tuned, e.g., how do the authors handle one-channel X-ray input; Is the text module also finetuned?
>
> The CXRs are converted to 3-channels and then the SD v1.5 model is finetuned with CXRs and texts from the MIMIC-CXR dataset. The SD v1.5 model weights are obtained from Hugging Face and the finetuning is done using accelerate. The text module is also finetuned.
>
> > What's the input of the copied and trained ControlNet compared with your proposed GazeDiff since ControlNet also requires an image or 2D input.
>
> The reviewer is right in pointing out that the ControlNet requires an image as input. We show in Figure 2 that (I, G’) is fed to the trainable copy. We show the HVA edge maps that are fed in Figure 3 and Figure 7 in the Appendix. In summary, the input to GazeDiff is the HVA edge maps and the text conditions.
>
> > Experiments using only canny edge of the X-ray images instead of HVA. How would the performance of this be?
>
> Using only canny edge, for Cell dataset, the accuracy is 58.01 (GazeDiff:59.29, 2.16% improvement) and the F1 score is 71.21 (GazeDiff:72.08, 1.21% improvement) and for the CHN dataset the accuracy is 58.61 (GazeDiff:58.91, 0.51% improvement) and the F1 score is 70.41 (GazeDiff:70.69, 0.40% improvement).
>
> > Experiments of other condition methods if time allows.
>
> We show results of ControlNet with various radiomics filter maps as controls on 2 datasets. We are adding this as Table 7 in the Appendix. We observe that GazeDiff outperforms all the ControlNets trained with these different conditions.
>
> **Reference**:
>
> [1] Chambon, Pierre, et al. "Roentgen: Vision-language foundation model for chest x-ray generation." arXiv preprint arXiv:2211.12737 (2022).
>
> [2] Li, Alexander C., et al. "Your diffusion model is secretly a zero-shot classifier." Proceedings of the IEEE/CVF International Conference on Computer Vision. 2023.
>
> [3] Radford, Alec, et al. "Learning transferable visual models from natural language supervision." International conference on machine learning. PMLR, 2021.
>
> [4] Eslami, Sedigheh, Gerard de Melo, and Christoph Meinel. "Does clip benefit visual question answering in the medical domain as much as it does in the general domain?." arXiv preprint arXiv:2112.13906 (2021).

---

> > ### Comment · Reviewer_hvVX · 2024-03-26
> >
> > Thank you for the efforts during the rebuttals as well as your revisions to the paper! I'm happy with the rebuttals.

---

### Official Review · Reviewer_H7kp · 2024-02-27

**Confidence:** 4
**Preliminary Rating:** 4
**Final Rating:** 4

**Summary:**

This work presents a novel architecture that leverages radiologists’ eye gaze patterns as controls for text-to-image diffusion models for zero-shot classification. This model offers more than just image generation capabilities; the density estimates derived from the gaze-guided diffusion model can effectively improve zero-shot classification performance. With eye gaze patterns as conditional inputs to diffusion models, the generated medical images contain disease-specific features or irregularities in the exact location or periphery of radiologists’ eye gaze fixations.

**Strengths:**

1. Medical information is used to assist image generation.
2. Experimental results show the model using gaze information can improve the performance. Especially the gaze information can guide the generated images to have the right disease position.

**Weaknesses:**

1. The experimental results lack evaluation from radiologists. The involvement of doctors in the evaluation of generated images can make the experiment more convincing.
2. The zero-shot classification lacks comparison with other methods.

**Detailed Comments:**

1. In Fig.4, "We show the generated pathology/object mentioned in the radiologists’ transcript in the red box.". Are these bounding boxes marked by doctors? Similarly, In Fig.5, are these bounding boxes also marked by doctors? Why are these bounding boxes not marked by other methods?
2. Can FID and CLIP scores better evaluate medical images? Are doctors involved in the evaluation of the generated images? If so, how are they evaluated?
3. It would be better to show the results of zero-shot classification compared with other methods.

**Justification Of Final Rating:**

This paper uses the gaze information to control the chest X-ray image generation. This is a good idea to control the disease location in the generated image. The generated image is expected to have higher quality compared with previous methods. I think this paper deserves accepted.

**Justification Of The Preliminary Rating:**

This work presents a novel architecture that leverages radiologists’ eye gaze patterns as controls for text-to-image diffusion models for zero-shot classification. With eye gaze patterns as conditional inputs, the generated medical images contain disease-specific features or irregularities in the exact location or periphery of radiologists’ eye gaze fixations. The involvement of doctors in the evaluation of generated images can make the experiment more convincing.

**Questions To Address In The Rebuttal:**

1. Introducing the doctor's evaluation work in the generated pictures. Due to the particularity of CXR pictures, the doctor's evaluation of the pictures is more in line with medical standards.
2. Compare the disease conditions in the pictures generated by different methods. Can these generated images pass disease diagnostic models and doctors’ inspections?
3. It would be better to show the results of zero-shot classification compared with other methods.

---

> ### Author Response · Authors · 2024-03-17
>
> We thank the reviewer for the excellent comments on our work. We answer the questions/comments below:
>
> > In Fig.4, "We show the generated pathology/object mentioned in the radiologists’ transcript in the red box.". Are these bounding boxes marked by doctors? Similarly, In Fig.5, are these bounding boxes also marked by doctors? Why are these bounding boxes not marked by other methods?
>
> Yes, in Figure 4 and 5, the bounding boxes and arrows are marked by a radiologist (mentioned in Subsection 3.4 in the paper.)
>
> > Can FID and CLIP scores better evaluate medical images? Are doctors involved in the evaluation of the generated images? If so, how are they evaluated?
>
> FID and CLIP scores are standard metrics for evaluating the quality of generated images. Also, in [1], FID and CLIP have been used for quantitative assessment of image fidelity and diversity.
>
> > Introducing the doctor's evaluation work in the generated pictures. Due to the particularity of CXR pictures, the doctor's evaluation of the pictures is more in line with medical standards.
>
> The reviewer points out a very interesting evaluation mechanism. The quality and clinical accuracy of the generated CXRs should be evaluated by a radiologist. In this work, in Figure 4 and Figure 5, we show the radiologist's annotation using red arrows or bounding boxes. The feedback of the radiologist indicated that the CXRs generated by the ControlNet model are of sub-optimal quality whereas the CXRs generated by GazeDiff are of high-quality and clinically accurate.
>
> > Compare the disease conditions in the pictures generated by different methods. Can these generated images pass disease diagnostic models and doctors’ inspections?
>
> Initial examination by an experienced radiologist suggests that the CXRs generated from GazeDiff are of good quality and diagnostic accuracy. We also showed CXRs generated by the ControlNet to the radiologist, the feedback was that the generated images using the ControlNet model were of poor quality (Figure 5).
>
> > It would be better to show the results of zero-shot classification compared with other methods.
>
> In Table 1 of  [2], CLIP has been used as another baseline method.  We have added CLIP and PubMedCLIP as  additional methods for comparison. See Table 1.
>
> **Reference**
>
> [1] Chambon, Pierre, et al. "Roentgen: Vision-language foundation model for chest x-ray generation." arXiv preprint arXiv:2211.12737 (2022).
>
> [2] Li, Alexander C., et al. "Your diffusion model is secretly a zero-shot classifier." Proceedings of the IEEE/CVF International Conference on Computer Vision. 2023.

---

> > ### Comment · Reviewer_H7kp · 2024-03-26
> >
> > Thanks for your clarification. I have no further questions.

---

### Official Review · Reviewer_UbDv · 2024-03-03

**Confidence:** 3
**Preliminary Rating:** 3
**Recommendation:** Poster
**Final Rating:** 4

**Summary:**

Overall, this is an interesting paper that presents a new approach to integrate expert visual attentions as an additional control to the text-to-image diffusion models. Four public domain datasets were used to evaluate the performance of the proposed GazeDiff. Given the common medical scenarios radiologists’ eye gaze patterns and transcripts are not available; the authors used zero-shot classification.

**Strengths:**

•	The architecture of the new approach (GazeDiff) is well explained. It consists of three major components: the Stable Diffusion (SD-CXR) model, the CN-Gaze model, and the class conditioned density estimates.

•	The proposed GazeDiff is compared against SD, ControlNet trained on natural images, and the same models finetuned on CXRs.

•	Overall, well-written paper with clear structure.

**Weaknesses:**

•	The key contributions of the proposed approach are not clearly outlined.

•	The list of datasets is provided, but there is no description accompanying each dataset.

•	In section 3.1, the justifications of selecting parameters to tune the model are not provided.

•	The results show marginal improvements which make me question the effectiveness and significance of the proposed approach.

**Detailed Comments:**

See the comments above.

**Justification Of Final Rating:**

The authors have addressed all concerns during the rebuttal process. The dataset description as well as the effectiveness and significance of the proposed approach are provided. I'm happy with the quality of the rebuttals.

**Justification Of The Preliminary Rating:**

See detailed comments in the weaknesses section. Addressing the points in the 'Questions To Address In The Rebuttal' will improve the quality of the paper and evidence the validity of the proposed approach.

**Questions To Address In The Rebuttal:**

•	Provide the key contributions in the introduction section.

•	A detailed description of datasets with samples.

•	The results show marginal improvements which make me question the effectiveness and significance of the proposed approach. In Table 1, how the performance of GazeDiff compares to other methods in terms of statistical significance.

---

> ### Author Response · Authors · 2024-03-17
>
> We thank the reviewer for the excellent comments on our work. We answer the questions/comments below:
>
> > In section 3.1, the justifications of selecting parameters to tune the model are not provided.
>
> We apologize we could not provide a detailed description on the parameter selection due to space constraints. Here, we elaborate on the parameters used for finetuning our models. For finetuning the Stable Diffusion v1.5 model, we use the same image size of 512x512 and learning rate of 1e-5 that was used to train the model on natural images in HuggingFace. We use a batch size of 4,the training was done on a Quadro RTX 8000 (48 GB). For finetuning the ControlNet model, we use the same parameters as in HuggingFace, the only difference is that we train for 10 epochs as the eye gaze dataset is comparatively small and requires more iterations to converge well.
>
> > Provide the key contributions in the introduction section.
>
> We have added key contributions in the introduction section.
> ‘The key contributions of this paper are as follows: a) we propose to add eye gaze patterns of experts as additional control and radiologists' text as prompt to the text-to-image diffusion models. b) we use this finetuned diffusion model as a zero-shot classifier for downstream pulmonary diseases like pneumonia and tuberculosis classification tasks.’
>
> > A detailed description of datasets with samples.
>
> We have added detailed description of the datasets with image samples used in Appendix A.
>
> > The results show marginal improvements which make me question the effectiveness and significance of the proposed approach. In Table 1, how the performance of GazeDiff compares to other methods in terms of statistical significance.
>
> Though the difference in accuracy and F1 is marginal. We show the statistical significance by calculating the paired t-test between the predictor errors ($\epsilon_\theta$) for 2 datasets. For the Cell Pneumonia dataset, the p value is 0.0067 and for the RSNA Pneumonia dataset, the p value is 0.0317.

---

> > ### Comment · Reviewer_UbDv · 2024-03-26
> >
> > Thank you for your efforts during the rebuttal process and for your revisions to the paper I'm pleased with the quality of the rebuttals.

---

### Official Review · Reviewer_JKPa · 2024-03-04

**Confidence:** 2
**Preliminary Rating:** 4
**Recommendation:** Poster

**Summary:**

This paper develops the GazeDiff model that utilizes the radiologist's eye gaze patterns as additional control of the text-to-image diffusion model. And show that this model can not only improve image generation capability, but also improve the zero-shot classification performance.

**Strengths:**

* The paper explores an interesting research topic of integrating human attention into the model and improving the model’s performance.
* Experiments are comprehensive.
* Paper is well-written, concepts and methods are introduced clearly.

**Weaknesses:**

* The authors explored a way to integrate human attention into the model. However, the supurior of the proposed model compare with foundation model (e.g.CheXagent) which is trained on large number of chest x-ray images and reports is not very clear. Especially considering that the inference time of diffusion model may hinder its application as a classifier.
* Since the generated image based on input and text are quite arbitrary and very different from the input (see  Figure.4), it makes the motivation of using the guided diffusion model as a zero-shot classifier questionable (diffusion model may perform classification by adding arbitrary information that may not exist in the input), especially in the high-risk medical domain.
* It is not clear why the authors use HVA edge maps instead of directly using HVA maps as control.

**Detailed Comments:**

Interesting work that explores how to use human expert knowledge to improve ML learning models. However, the application scenario and the tasks are not very convincing.

**Justification Of The Preliminary Rating:**

The research topic this paper explores is interesting and potentially inspires other research in utilizing human expertise in guiding or controlling the ML model. However, the zero-short classification task the model evaluated on is not very convincing and not very suitable for the application in medical domain.

**Questions To Address In The Rebuttal:**

Please address the Weaknesses points

---

> ### Author Response · Authors · 2024-03-17
>
> We thank the reviewer for the excellent comments on our work. We answer the questions/comments below:
>
>  > The authors explored a way to integrate human attention into the model. However, the supurior of the proposed model compare with foundation model (e.g.CheXagent) which is trained on large number of chest x-ray images and reports is not very clear. Especially considering that the inference time of diffusion model may hinder its application as a classifier.
>
> The reviewer rightly points out that CheXagent is trained on a large number of CXR images and reports from MIMIC-CXR, PadChest, and BIMCV-COVID-19 and demonstrates superior performance for multiple tasks. The main difference between foundation models like CheXagent and diffusion models is that diffusion models demonstrate excellent zero-shot performance on unseen tasks. Finetuning CheXagent, a model with 8 billion parameters with eye-gaze of radiologists, is challenging. Moreover, the probabilistic nature of diffusion models enables them to capture intricate data distributions, in our case visual attention patterns, more effectively than foundation models.
> The reviewer points to an interesting concern regarding inference time of diffusion models. Diffusion models are indeed comparatively slower during inference but the advantages outweighs this limitation. This can be addressed in the mentioned points:
> Model Complexity: In clinical scenarios, generalization to unseen tasks and zero-shot inference is of prime importance. Existing zero-shot classification methods typically involve more data and complex inference processes whereas the architecture and the probabilistic inference mechanisms of the diffusion model helps in better generalization to unseen classes with very less data.
> Real-time clinical scenarios:  In clinical scenarios, where the training data is often sparse and, in cases unavailable, diffusion models demonstrate superior ability to generalize to unseen classes without retraining the model. Diffusion models can generate high quality medical images that also help in diagnosis.
>
> > Since the generated image based on input and text are quite arbitrary and very different from the input (see Figure.4), it makes the motivation of using the guided diffusion model as a zero-shot classifier questionable (diffusion model may perform classification by adding arbitrary information that may not exist in the input), especially in the high-risk medical domain.
>
> GazeDiff generates CXRs with the disease pathology in Figure 4 verified by an experienced radiologist. The reviewer raises a concern on the ambiguity between the text and the generated CXR. This is a critical problem in text-to-image diffusion models. To address this, we propose a HVA guided CXR generation using ControlNet (CN-Gaze). This helps in generating high-quality and clinically accurate CXR images by integrating the visual attention information from radiologists. We are using this CN-Gaze architecture as a zero-shot classifier (GazeDiff). Then, we show that GazeDiff performs better in zero-shot classification compared to CN-CXR and SD-CXR. This implies that GazeDiff generates better disease patterns than the baselines. Hence, the applicability of these types of conditional diffusion models in data augmentation for classification can be useful.
> Concern in high-risk medical domains: The arbitrary nature of the images generated from a diffusion model can be risky if it is used in real-time clinical practice. In recent times, diffusion models are used in improving performance for diagnostic tasks like disease classification, segmentation and zero-shot inference. Here, the main application is to generate images that have the mentioned disease pathology and are of diagnostic quality.
>
> > It is not clear why the authors use HVA edge maps instead of directly using HVA maps as control.
>
> In this work, we use HVA edge maps rather than using HVA maps as the former is a strong control that helps in reducing ambiguity between the control and the generated CXRs. HVA maps are heatmaps that are visual representations of eye gaze fixation distribution across a spatial domain and hence sparsifies the control resulting in ambiguity whereas HVA edge maps are edge maps obtained from applying Canny edge filtering to the HVA map that contain information regarding both the physical anatomy and disease presence information - this minimizes the ambiguity and acts as a strong control.
>
> > Interesting work that explores how to use human expert knowledge to improve ML learning models. However, the application scenario and the tasks are not very convincing.
>
> The method outlined offers a broad range of applications, spanning from unseen task generalization to missing modality generalization and time point image generation, among others. Also, finetuning diffusion models with radiologist’s eye gaze attention maps have immense potential towards generating clinically accurate medical images.

---

### Comment · Area_Chair_7yzm · 2024-03-17
**Please read and respond to author comments**

Dear reviewers, The authors have posted responses to your reviews. Please take the time to read and respond before March 27.

---

> ### Author Response · Authors · 2024-03-26
> **Request for feedback**
>
> We express our gratitude to the reviewers again for their comprehensive assessment during the initial round and eagerly await any additional feedback to address potential clarifications in our responses.

---

### Author Response · Authors · 2024-03-17

We thank the reviewers for their insightful comments and suggestions on our work which really helped improve our paper. The reviewers point out that integrating the eye gaze patterns of radiologists to a diffusion model is interesting and helps in improving the zero-shot classification performance. The reviewers also appreciate the structure and clarity of the paper. However, the reviewers have a few concerns about comparison with other methods, improvements in the performance, comparison with other conditions, etc. We have addressed all these concerns in this rebuttal in detail. Also, the reviewers had a few comments requiring additional information, we have addressed them and modified the paper likewise.

---

### Author Response · Authors · 2024-03-26
**No feedback yet**

Dear Area Chair,

We have not received any response to the rebuttal yet despite your earlier reminder. We are concerned that this delay may not give us adequate time for further experimentation should any comments be forthcoming today. We have posted another reminder.

Regards,
Authors of Paper #73

---

### Meta-Review · Area_Chair_7yzm · 2024-04-04

**Recommendation:** Accept (Poster)
**Confidence:** 5

**Metareview:**

This paper proposes a novel text-to-image diffusion models for zero-shot classification of CXRs that also uses radiologists' gaze for conditioning. In general the ideas was thought to be interesting and the use of more medical information in image generation was thought to be a strength. The reviewers had some questions about contributions of the work, minor improvements against methods compared against and lack of evaluation by radiologists, these appear to have been resolved in the rebuttal. Even though the improvements are minor, it was pointed out that they are statistically significant. Some reassurance from radiologist feedback was also provided in the rebuttal. There was some remaining questions by one reviewer about limited novelty, all reviewers leaned towards acceptance of the paper in their final ratings.

---

### Decision · Program_Chairs · 2024-04-05

Accept (Poster)